# Selection and Validation of Reference Genes for Gene Expression Studies in *Euonymus japonicus* Based on RNA Sequencing

**DOI:** 10.3390/genes15010131

**Published:** 2024-01-21

**Authors:** Wei Guo, Yihui Yang, Bo Ma, Wenbo Wang, Zenghui Hu, Pingsheng Leng

**Affiliations:** 1College of Landscape Architecture, Beijing University of Agriculture, Beijing 102206, China; 202040240002@bua.edu.cn (W.G.); 202230512017@bua.edu.cn (B.M.); 20239401@bua.edu.cn (W.W.); 2Engineering Research Center for Ancient Tree Health and Ancient Tree Culture of National Forestry and Grassland Administration, Beijing 102206, China

**Keywords:** *E. japonicus*, RNA-seq, traditional housekeeping genes, new reference genes, qPT-PCR

## Abstract

*Euonymus japonicus* is one of the most low-temperature-tolerant evergreen broad-leaved tree species in the world and is widely used in urban greening. However, there are very few molecular biology studies on its low-temperature tolerance mechanism. So far, no researcher has selected and reported on its reference genes. In this study, 21 candidate reference genes (12 traditional housekeeping genes and 9 other genes) were initially selected based on gene expression and coefficient of variation (CV) through RNA-Seq (unpublished data), and qRT-PCR was used to detect the expression levels of candidate reference genes in three different groups of samples (leaves under different temperature stresses, leaves of plants at different growth stages, and different organs). After further evaluating the expression stability of these genes using geNorm, NormFinder, Bestkeeper, and RefFind, the results show that the traditional housekeeping gene *eIF5A* and the new reference gene *RTNLB1* have good stability in the three different groups of samples, so they are reference genes with universality. In addition, we used *eIF5A* and *RTNLB1* as reference genes to calibrate the expression pattern of the target gene *EjMAH1*, which confirmed this view. This article is the first to select and report on the reference gene of *E. japonicus*, laying the foundation for its low-temperature tolerance mechanism and other molecular biology research.

## 1. Introduction

*E. japonicus* is a shrub of the genus *Euonymus* that is evergreen all year round, is extremely resistant to pruning and shaping, and has significant biotic and abiotic resistance. It can tolerate low temperatures of −28.5 °C and is, therefore, suitable as a model plant to study the molecular mechanisms of plants under freezing stress [1]. In addition, because of its high ornamental and economic value and its strong adaptability, *E. japonicus* is very popular in garden styling and landscaping in many parts of the world. *E. japonicus* is also an important medicinal plant. The natural chemical components in its body not only have anti-tumor and blood stasis effects, but also have insecticidal, bactericidal, and anti-viral activities and can be developed into drugs for biological control. Since the 1980s, scholars have conducted extensive research on the reproductive technology, pest control, stress resistance biology, and other physiological aspects of *E. japonicus* [2,3,4]. However, regarding its molecular biology and gene expression, research is very scarce. Currently, there are only 130 genes, 729 protein sequences, and zero SNP markers published in the NCBI. There is a serious lack of information on the entire genome and molecular markers of *E. japonicus*. In addition, no researchers have yet selected and reported on the reference genes (RGs) of *E. japonicus*, which greatly limits its molecular biology research.

qRT-PCR has the characteristics of extremely high sensitivity, specificity, and repeatability and high cost performance [5]. To ensure the accuracy and reliability of qRT-PCR results, it is crucial to select appropriate RGs to normalize the expression levels of genes. The ideal RGs should have moderate and stable expression levels in different developmental stages, different experimental treatments, and different organs [6,7,8]. Examples of RGs include the commonly used *18S rRNA*, actin (*ACT*), tubulin (*TUB*), eukaryotic translation initiation factor (*eiF*), and glyceraldehyde-3-phosphate dehydrogenase (*GAPDH*) [9,10,11,12,13]. However, studies have shown that these genes are not always stable in different species or under different experimental conditions [14,15,16]. In addition, the selection of RGs is a time-consuming and labor-intensive process. Therefore, it is very meaningful and necessary to select universal RGs for a specific species to meet different experimental needs.

Due to the lack of gene sequence and expression information, the selection of RGs of *E. japonicus* has not been carried out so far, and RNA-seq can effectively solve this problem. A transcriptome is a collection of RNA at a specific time point in organisms under different growth conditions [17]. RNA-seq can comprehensively and quickly obtain all of the transcript information in a sample. It is an effective way to mine genetic resources and especially suitable for plants without reference genomes [18,19]. This also means that the selection of RGs is no longer limited to traditional housekeeping genes, providing the possibility to select new stable RGs. Nowadays, RNA-seq has been widely used in the selection of RGs [20,21,22,23]. Therefore, the purpose of this study was to use RNA-seq to select and verify universal RGs that are suitable for *E. japonicus*.

This study used leaves of *E. japonicus* at different overwintering times as materials. Based on the results of RNA-seq, 11 candidate traditional housekeeping genes and 9 candidate new RGs were initially selected, and their expression stability was then evaluated using geNorm, NormFinder, Bestkeeper, and RefFinder. Through calculations and analysis, among the candidate traditional housekeeping genes, *eIF5A* was found to be the most stably expressed in leaves under different temperature stresses, leaves of plants at different growth stages, and different organs, and among the nine candidate new RGs, *RTNLB1* was found to be the most stably expressed. The stability of the two RGs was further verified through qRT-PCR, and we found that they can be used alone or in combination. Our study is the first to select and report on the RGs of *E. japonicus*, laying the foundation for its low-temperature tolerance mechanism and other molecular biology research.

## 2. Materials and Methods

### 2.1. Plant Materials

The materials used for RNA-seq were the leaves (second to fourth leaf position) of perennial (more than 15 years old) *E. japonicus* from the south bank of Shahe Reservoir in Changping District, Beijing during the natural overwintering process. The collection times were 6 November, 8 December, and 8 January 2020 and 6 February and 5 March 2021, for a total of 5 samples. The materials used for qRT-PCR were divided into three groups: leaves under different temperature stresses; leaves of plants at different growth stages; and different organs. The isolated branches and leaves of perennial *E. japonicus* from the south bank of Shahe Reservoir were stored at 25 °C, 5 °C, 0 °C, −5 °C, −10 °C, and −15 °C for 24 h (a period of 12 h in the light (300 μmol·m^−2^·s^−1^) and 12 h in the dark) with a humidity of 60%. After the treatments, the leaves were collected as the samples. Leaves of plants at different growth stages (4 months, 1 year, 2 years, 5 years, and greater than 10 years) were collected from Beijing University of Agriculture, and the age of each plant was based on the age when it was planted at Beijing University of Agriculture plus the growth cycle after planting to the time of sampling. Different organs (roots, stems, leaves, buds, and fruits) were collected from biennial *E. japonicus* of the same plant at Beijing University of Agriculture. Three biological replicates were set up for each material.

### 2.2. RNA-Seq

RNA-seq was completed by BGI using the DNBSEQ sequencing platform for leaves in 5 different overwintering periods. Three biological replicates were set for each period, resulting in a total of 15 sequencing samples. The sequencing method used was RNAdenovo (a type of RNA-seq suitable for species without a reference genome), the fragment length was 100–500 bp, the sequencing mode was PE150, and the total data volume was 6 G bp. The specific process was as follows. First, the total RNA of the sample was extracted, and then the mRNA library (DNBSEQ) was constructed and sequenced. The raw data obtained by sequencing were filtered using the filtering software SOAPnuke (v1.6.5) [24] to obtain clean data. Then, we used Trinity (v2.13.2) [25] to de novo assemble the clean reads, and we used Bowtie2 (v2.4.5) [26] software to align the obtained clean data to the reference gene sequence (Unigenes obtained by de novo assembly). Then, RSEM (v1.3.1) [27] was used to calculate the expression levels of genes and transcripts as fragments per kilobase of transcript per million mapped reads (FPKM), and the assembled Unigene was annotated with NR databases (https://ftp.ncbi.nlm.nih.gov/blast/db/, accessed on 24 May 2021).

### 2.3. RNA Extraction and cDNA Reverse Transcription

RNA extraction was performed using an EASYspin Plus Complex Plant RNA Kit (Aidlab, Beijing, China). We used 1% agarose gel electrophoresis to assess the integrity of the RNA. The bands were clear, and the brightness of 28S at about 1000 bp was about twice that of 18S at about 600 bp (Figure 1), indicating that the integrity of the RNA was good. The RNA samples were then quantified using NanoPhotometer P330 (Imlen, Germany). The A260/A280 of qualified RNA must be in the range of 1.8–2.2, and A260/A230 > 2.0. Only RNA that met the criteria was used for subsequent analyses. cDNA Synthesis SuperMix (TransGen, Beijing, China) was used to reverse-transcribe 400 ng of total RNA from each sample into cDNA.

### 2.4. Primer Design and PCR

Based on the sequences selected from the RNA-seq data on *E. japonicus*, primers were obtained using Primer-BLAST of the NCBI. The primer design standards were as follows: length, 18–23 bp; GC content, 40–60%; melting temperature, 58–62 °C; amplicon length, 90–200 bp. The designed primer pairs were aligned back to the gene sequence file obtained by RNA-seq to ensure that only the target genes could theoretically be amplified. Conventional PCR was completed using S1000 (Bio-Rad, Hercules, America), and the system and program (Tm = 60 °C) were configured according to the instructions of the Ape x HF HS DNA Polymerase FS kit (Accurate, Changsha, China). qPT-PCR was completed using CFX96 (Bio-Rad, America), and the system and procedures were configured according to the instructions of the PerfectStart^®^ Green qPCR SuperMix kit (TransGen, China). After amplification, melting curve analysis was performed, and the amplification efficiency was calculated according to E = [10 (−1/slope) − 1] × 100%. In the qPT-PCR, in order to verify the primer specificity, amplification efficiency, and expression profile, the template used was a cDNA mixture (5 μL of each sample from three groups (as described in Section 2.1)). First, the mixed cDNA was diluted to 500 ng/μL, and then it was diluted into 6 gradients (2^0^, 2^−1^, 2^−2^, 2^−3^, 2^−4^, and 2^−5^) as templates. Each qPT-PCR reaction was set up with 3 technical replicates.

### 2.5. Data Analysis

Four commonly used methods, geNorm [28], NormFinder [29], BestKeeper [30], and RefFinder [31,32], were used to analyze the expression stability of candidate RGs. All four methods can be applied on the web page https://blooge.cn/RefFinder/, accessed on 3 July 2022. The specific method is as follows. First, we opened the webpage https://blooge.cn/RefFinder/ accessed on 3 July 2022, and then entered the qPT-PCR results into the indicated text box. The entered data must meet some format requirements, which can be obtained by clicking the “Try example” button below the text box. After editing the qPT-PCR data as required and inputting them into the text box, we clicked the “Analyze” button below the text box to obtain the stable values and ranking results of the candidate RGs under the four algorithms. In addition, IBM SPSS 22 was used for analysis of variance, and origin 2018 was used for data visualization.

## 3. Results

### 3.1. Screening of Candidate RGs Based on the Results of RNA-Seq

The RGs not only require stable expression, but also a high expression level because low-expressed genes are difficult to detect and quantify [21]. Referring to the relevant studies [33,34], we used average FPKM > 50 and CV < 0.2 as standards to eliminate genes in the transcriptome and screened out 12 candidate traditional housekeeping genes (Table 1). Then, the other genes in the transcriptome were ranked from small to large according to the CV, and the top nine were selected as candidate new RGs (Table 1). Through blastn of the NCBI, the identities of 21 candidate RGs were determined, and their homology with *Tripterygium wilfordii*, a member of the same family, was over 88% (Table 1). The FPKM and CV of candidate RGs are listed in Appendix A.

### 3.2. Primer Specificity and Amplification Efficiency of Candidate RGs

First, the NCBI’s Primer-BLAST was used to obtain the primers for candidate RGs (Table 2). We then performed ordinary PCR amplification and used 1.5% agarose gel electrophoresis to detect the PCR products. Their sizes were between 90 and 200 bp (Figure 2), which was consistent with the expected results (Table 2), and the band in each lane was single and bright, indicating that the primers have strong specificity. Then, qRT-PCR was performed using six concentration gradient cDNA mixtures as templates, and a standard curve was drawn based on the results. We determined that the amplification efficiency of 21 genes was between 98% and 110%, indicating that the amplification efficiency was qualified. The linear correlation coefficient (R^2^) of the standard curves was between 0.998 and 1.000 (Table 2), indicating that the results are reliable. In addition, the melting curves of 21 genes all showed a single melting peak between 80 and 85 °C, further proving the specificity of the primers (Figure 3).

### 3.3. Expression Profiles of Candidate RGs

By using the cDNA mixtures (as described in Section 2.1) as templates, qRT-PCR was performed on the qualified primers, and their expression abundanc and stability were analyzed. The threshold period (Ct value) of qRT-PCR can directly reflect the expression level of the gene, which is inversely proportional to the expression abundance. The average Ct value of the 12 candidate housekeeping genes in leaves under different temperature stresses, leaves of plants at different growth stages, and different organs ranged from 20 to 29, indicating a moderate level of expression. Among them, *UBQ*, *ClpA*, and *eIF5A* had a higher expression abundance, and *RAP2* and *UBC2* had a lower expression abundance. Judging from the dispersion of the Ct value distribution, *eIF5A* is the most stable one (Figure 4a). The expression level of the nine candidate new RGs was also moderate, with average Ct values between 24 and 27. The difference in expression abundance among them was small. In addition, *HMGB2* and *RTNLB1* had the higher expression abundance, while *UNC* and *SCPL* had the lower expression abundance (Figure 4b). By comparison, it can be seen that the distribution of the average Ct values (24–27) of candidate new RGs is more concentrated than that of the candidate housekeeping genes (20–29), indicating that the difference in expression abundance between housekeeping genes is greater.

### 3.4. GeNorm Analysis

GeNorm usually selects the two most suitable candidate genes as RGs. By calculating the standard deviation of the candidate RGs, the stability value M is obtained, and then the candidate RGs are ranked according to the size of M (the smaller the value, the more stable the gene) [28]. According to the GeNorm analysis results (Figure 5), in leaves under different temperature stresses, leaves of plants at different growth stages, and different organs, the most stable housekeeping genes are *eIF5A* and *UBC2*, *TUA* and *TUB*, and *Actin7* and *TUA,* respectively (Figure 5A–C). Among these three groups of samples, *TUA*, *eIF5A,* and *TUB* ranked highly and had little change in ranking. Among the new RGs, the most stable genes are *UNC* and *GEPI48*, *RTNLB1* and *GAD*, and *SBT3.17* and *SCPL,* respectively. The ones with smaller stability changes and a higher ranking are UNC and *RTNLB1*.

### 3.5. NormFinder Analysis

NormFinder can directly evaluate the stability of RGs based on variance analysis and rank them according to the stability value. It considers the gene with the smallest stability value to be the most stable [35]. According to NormFinder, the most stable ones in leaves under different temperature stresses, leaves of plants at different growth stages, and different organs are *TUA* and *eIF5A*, *eIF5A* and *TUB*, and *eIF5A* and *GAPDH,* respectively. The stability rankings of *eIF5A* and *GAPDH* among the three groups of samples were high, while the rankings of *TUA* and *TUB* were not stable (Table 3). Among the nine new RGs, the top ones are *SCPL* and *RTNLB1*, *ALEU* and *HMGB2*, and *HMGB2* and *RTNLB1,* respectively. Taken together, *RTNLB1* ranks the highest in stability among the three groups of samples, and *ALEU* and *HMGB2* have larger changes in ranking (Table 3).

### 3.6. Bestkeeper Analysis

BestKeeper evaluates the stability of genes by calculating the SD and CV. The smaller the SD and CV values, the higher the stability, and, when SD > 1, the expression of this gene is unstable [36]. The housekeeping genes with the highest stability in leaves under different temperature stresses, leaves of plants at different growth stages, and different organs are *eIF5A* and *UBC2*, *TUB* and *GAPDH*, and *Actin7* and *eIF5A,* respectively. The SD of *Actin7* is greater than 1 under different temperature stresses (Table 4). Similarly, *FtsH2*, *ClpA*, *RAP2*, *UBC2*, *CYP72A*, etc. are not suitable for use as RGs. Among the nine candidate new RGs, the ones with the highest stability rankings are *HMGB2* and *SBT3.17*, *HMGB2* and *RTNLB1*, and *SCPL* and *SBT3.17,* respectively. *GEPI48*, *UNC*, and *ALEU* are not suitable for use as RGs (Table 4).

### 3.7. Comprehensive Analysis of RefFinder

RefFinder assigns appropriate weights to the results of candidate RGs in the four algorithms geNorm, Normfinder, BestKeeper, and Delta-Ct, and then calculates the geometric mean of the stable value weights of all algorithms and performs overall ranking to achieve a comprehensive ranking [37]. The most stable housekeeping genes in leaves under different temperature stresses, leaves of plants at different growth stages, and different organs are *eIF5A* and *TUA*, *TUB* and *eIF5A*, and *eIF5A* and *Actin7,* respectively. Among them, *eIF5A* ranks the highest in stability among the three groups of samples (Table 5). Among the nine new RGs, the top ones were *SCPL* and *RTNLB1*, *ALEU* and *RTNLB1*, and *HMGB2* and *SCPL*, respectively. Among them, the stability of *ALEU* ranked last in leaves treated with different temperatures and in different organs, the stability of *SCPL* ranked sixth in leaves of plants at different growth stages, and *RTNLB1* ranked fourth in different organs (Table 5). Therefore, in contrast, *RTNLB1* is more universal as a reference gene.

### 3.8. Reference Gene Selection and Validation

The three algorithms GeNorm, NormFinder, and BestKeeper have both similarities and differences in principle, and their stability ranking of genes is also the same, which is not difficult to see from the results. Through a comparison, it was found that *eIF5A* ranked highly in leaves under different temperature stresses, leaves of plants at different growth stages, and different organs, and consistent results were obtained in RefFinder’s comprehensive algorithm. Therefore, we speculate that *eIF5A* is a high-quality reference gene with greater universality. The rankings of some candidate new RGs changed greatly in the three groups of samples. For example, in the results obtained by GeNorm, *GEPI48* ranked highly in leaves under different temperature stresses but had a low ranking in leaves of plants at different growth stages and different organs. On the contrary, *RTNLB1* ranked highly in all three groups of samples, and it also obtained a good ranking in the RefFinder algorithm, so we believe that *RTNLB1* also has good universality.

In order to verify the applicability of *eIF5A* and *RTNLB1*, we used the structural gene *EjMAH1* of *E. japonicus* as the target gene to calibrate its expression pattern. In our previous studies, we found that the wax content of the leaf epidermis is likely to play an important role in the natural overwintering process of E. japonicus, and EjMAH1 is one of the key regulatory genes in the wax synthesis pathway. According to the RNA-seq data, its expression levels are significantly differentially expressed during the natural overwintering process, which aroused our interest in EjMAH1’s expression pattern. When using *eIF5A* or *RTNLB1* or a combination of the two (*eIF5A* + *RTNLB1*) as the internal control, the relative expression of *EjMAH1* in leaves under different temperature stresses showed a downward trend, and the expression levels were all at their lowest at 0 °C (Figure 6a). In leaves of plants at different growth stages, all showed an overall trend of first increasing and then decreasing, and they were all at their highest in 2-year-old leaves (Figure 6b). In different organs, the expression levels all increased first and then decreased, and the genes were not expressed in roots or fruits (Figure 6c). It can be seen that, in the three groups of samples, when using *eIF5A* or *RTNLB1* as the reference gene, *EjMAH1* can obtain almost consistent expression patterns and, when using a combination of the two as the reference gene, consistent results can still be obtained, which further proves the reliability of *eIF5A* and *RTNLB1*.

## 4. Discussion

At present, there are few studies on the molecular biology of *E. japonicus*, and there are no reports on *E. japonicus* RGs, which are indispensable reference conditions for the study of gene transcription and regulation. Therefore, screening suitable RGs for *E. japonicus* is of great importance for subsequent molecular biology studies. RNA-seq can not only provide gene sequence information, but also output its expression level (as an FPKM value). The coefficient of variation (CV) can be obtained through the FPKM value of the gene in different samples. The smaller the CV, the better the stability. Therefore, the CV value can be used to initially screen out stable candidate RGs from the transcriptome. In other words, in addition to focusing on traditional housekeeping genes, it is also possible to screen other more stable new RGs, which is also a major advantage of RNA-seq. Traditional housekeeping genes such as *ACT*, *TUB*, and *eIF4A* have been proven to be qualified RGs [20,38,39]. Therefore, such genes are likely to also be applicable to *E. japonicus*. In addition, researchers have used RNA-seq to screen out new high-quality RGs, such as *SAP5* and *UXS3,* for *Brassica napus* [34], indicating that RNA-seq is an effective way to screen new RGs for specific species.

In our study, 12 candidate traditional housekeeping genes and 9 candidate new RGs were initially screened based on the transcriptome data on *E. japonicus*. These genes are moderately expressed in leaves under different temperature stresses, leaves of plants at different growth stages, and different organs. Among the candidate traditional RGs, the expression abundance of *UBQ*, *ClpA*, and *eIF5A* is higher, and the Ct value distribution of *eIF5A* is more concentrated, making it more suitable as a reference gene. The differences in expression abundance and Ct value distribution range between candidate new RGs are small, among which *HMGB2* has a higher expression abundance and a more concentrated Ct value distribution. However, the expression profiles only showed the expression abundance and discreteness of the RGs and did not analyze and rank them, so it is not sufficient to evaluate the stability of the RGs [21]. Therefore, it is very important to conduct further analysis with the help of RG stability analysis software. Based on the GeNorm algorithm, the most stable ones in leaves under different temperature stresses, leaves of plants at different growth stages, and different organs are *eIF5A* and *UBC2*, *TUA* and *TUB*, and *Actin7* and *TUA,* respectively; in NormFinder, they are *TUA* and *eIF5A*, *eIF5A* and *TUB*, and *eIF5A* and *GAPDH*, respectively; and, in BestKeeper, they are *eIF5A* and *UBC2*, *TUB* and *GAPDH*, and *Actin7* and *eIF5A,* respectively. It can be seen that the optimal RGs screened by the three algorithms have both similarities and differences. Similar results were also obtained in research on *Populus trichocarpa* [40], *Betula glabra* [41], and other plants. This is caused by the fact that the principles of the three algorithms are not exactly the same. Therefore, in order to avoid the one-sidedness of a single piece of analysis software, it is very important to conduct a comprehensive analysis and evaluation.

RefFinder assigns appropriate weights to the results of candidate RGs in each program based on the GeNorm, NormFinder, BestKeeper, and Delta-Ct algorithms, and then performs stability ranking by calculating the geometric mean of the stability value weights of all algorithms. Because of its comprehensive analysis advantage, this method is increasingly being used in RG studies [42,43]. In our study, *eIF5A* ranked highly in leaves under different temperature stresses, leaves of plants at different growth stages, and different organs, and consistent results were obtained in RefFinder. Therefore, we speculate that *eIF5A* is a high-quality reference gene with greater universality in *E. japonicus*. Similarly, we believe that *RTNLB1* is an excellent new reference gene for *E. japonicus*. After selecting the RGs, we used *EjMAH1* as the target gene and confirmed that, in the three different groups of samples, when using *eIF5A* or *RTNLB1* or a combination of the two as the reference gene, the results obtained are in good agreement, which further proves the reliability of using *eIF5A* and *RTNLB1* as RGs of *E. japonicus*.

*eIF5A* is a small acidic polypeptide with a molecular weight of 17–21 kD that is present in all eukaryotes. It remains the only protein found to date that contains carboxyputrescine lysine (Hypusine) residues [44]. *eIF5A* is involved in the first stage of peptide bond formation during translation, and experimental evidence shows that it is a universally conserved translation elongation factor [45], so it can be used as an reference gene. Its presence has been confirmed in research on species such as *P. tomentosa* [13], *Iris. lactea* var. chinensis [46], *Radopholus similis* [47], and sweet orange (*Citrus sinensis*) [48]. Our study found that, among the 12 candidate traditional housekeeping genes in *E. japonicus*, *eIF5A* is the most suitable reference gene. Reticulins (RTNs) were originally described as integral membrane proteins in the endoplasmic reticulum (ER) of mammalian neurons [49]. They have been shown to be involved in endocytosis [50,51] and are thought to play a role in apoptosis, axonal growth, and regeneration. Homologs of reticulon-like proteins (RTNLs) are divided into six reticulon-like protein subfamilies, including a plant subfamily named RTNLB (*RTNLB1* is in this subfamily). Currently, little is known about the function of RTNLB, and it has not been proven to be a reference gene. However, among the nine candidate new RGs we selected, the stability analysis and verification results indicate that *RTNLB1* can be used as a reference gene of *E. japonicus*.

## 5. Conclusions

This study proved for the first time that *eIF5A* and *RTNLB1* can be used as RGs of *E. japonicus*, and qRT-PCR experiments were conducted in leaves under different temperature stresses, leaves of plants at different growth stages, and different organs. Therefore, when performing qRT-PCR on E. japonicus, researchers could consider using *eIF5A* or *RTNLB1* alone or a combination of the two as reference genes. This lays the foundation for gene expression and other related molecular biology research on *E. japonicus*. However, caution should be exercised when incorporating these two RGs into other qRT-PCR experiments, as their expression stability has only been verified in the above respects and may be affected by other experimental conditions.

## Figures and Tables

**Figure 1 genes-15-00131-f001:**
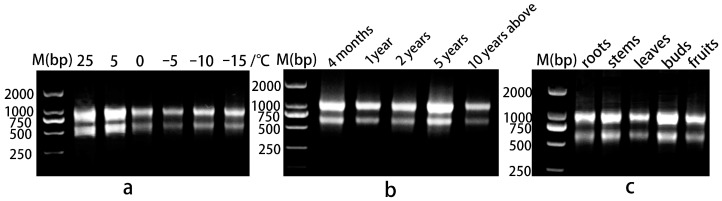
Agarose gel electrophoresis of RNA. (**a**) Leaves under different temperature stresses; (**b**) leaves of plants at different growth stages; (**c**) different organs.

**Figure 2 genes-15-00131-f002:**
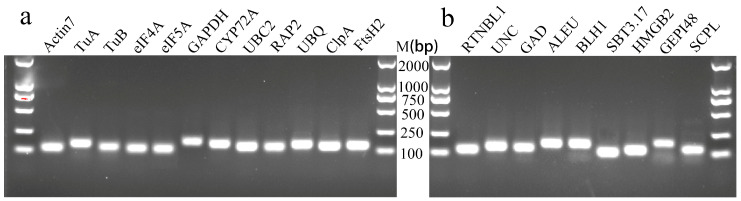
Agarose gel electrophoresis of PCR products of 21 candidate RGs. (**a**) The 12 candidate traditional housekeeping genes; (**b**) the 9 candidate new RGs.

**Figure 3 genes-15-00131-f003:**
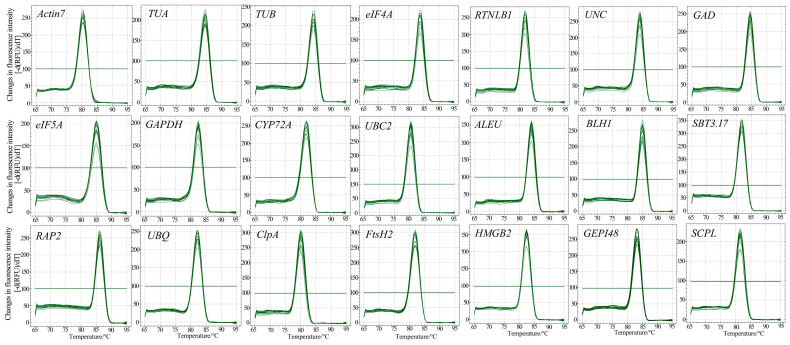
Specificity of primer pairs of qRT-PCR amplification. Melting curves of 21 candidate RGs exhibiting only single peaks.

**Figure 4 genes-15-00131-f004:**
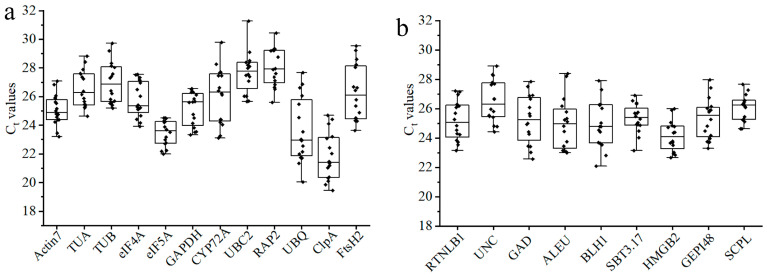
The expression profiles of candidate RGs across three different groups of samples. (**a**) The 12 candidate traditional housekeeping genes; (**b**) the 9 candidate new RGs. Each point in the box is the Ct value size, and the line across the box represents the median. The box points out the 25th and 75th percentiles. Whiskers represent the maximum and minimum values.

**Figure 5 genes-15-00131-f005:**
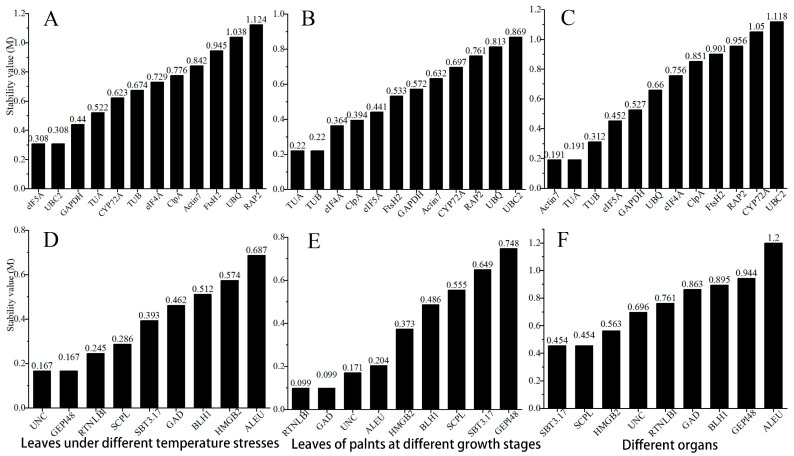
Stability values of candidate RGs analyzed by GeNorm. (**A**–**C**) Candidate traditional housekeeping genes in leaves under different temperature stresses, leaves of plants at different growth stages, and different organs, respectively; (**D**–**F**) Candidate new RGs in leaves under different temperature stresses, leaves of plants at different growth stages, and different organs, respectively.

**Figure 6 genes-15-00131-f006:**
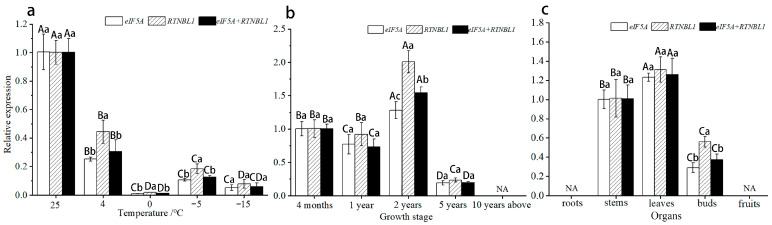
Expression patterns of *EjMAH1* in three groups of samples using *eIF5A*, *RTNLB1,* and a combination of these genes as reference genes. (**a**) Leaves under different temperature stresses; (**b**) leaves of plants at different growth stages; (**c**) different organs. NA indicates that no expression was detected. Note: Different capital letters indicate significant differences at 0.05 levels between treatments using the same RG, and different lowercase letters indicate significant differences at 0.05 levels between the same treatments using different RGs.

**Table 1 genes-15-00131-t001:** The information on 21 candidate RGs and NCBI homologous comparison results with *Tripterygium wilfordii*.

Category	Number	Gene	Gene Description	Homologous Sequence	Identities
Candidate traditional housekeeping genes	1	*Actin7*	actin-7	XM_038865290.1	96%
	2	*TuA*	tubulin α-5 chain	XM_038840525.1	96%
	3	*TuB*	tubulin β chain-like	XM_038823732.1	95%
	4	*eIF4A*	eukaryotic initiation factor 4A-3	XM_038845408.1	92%
	5	*eIF5A*	eukaryotic translation initiation factor 5A	XM_038856304.1	94%
	6	*GAPDH*	glyceraldehyde-3-phosphate dehydrogenase, cytosolic	XM_038838191.1	91%
	7	*CYP72A*	cytochrome P450 72A765	MN738192.1	90%
	8	*UBC2*	ubiquitin-conjugating enzyme E2 2	XM_038844679.1	95%
	9	*RAP2*	ethylene-responsive transcription factor RAP2-4	XM_038860091.1	90%
	10	*UBQ*	polyubiquitin	XM_038834479.1	91%
	11	*ClpA*	ATP-dependent Clp protease ATP-binding subunit ClpA	XM_038830531.1	96%
	12	*FtsH2*	ATP-dependent zinc metalloprotease FTSH 2	XM_038853108.1	94%
Candidate new RGs	1	*RTNLB1*	reticulon-like protein B1	XM_038853888.1	94%
	2	*UNC*	uncharacterized	XM_038867966.1	91%
	3	*GAD*	glutamate decarboxylase	XM_038864250.1	92%
	4	*ALEU*	thiol protease aleurain-like	XM_038855394.1	92%
	5	*BLH1*	BEL1-like homeodomain protein 1	XM_038829031.1	88%
	6	*SBT3.17*	subtilisin-like protease SBT3.17	XM_038863645.1	92%
	7	*HMGB2*	high mobility group B protein 2-like	XM_038849099.1	92%
	8	*GEPI48*	UDP-glucose 4-epimerase GEPI48-like	XM_038869380.1	96%
	9	*SCPL*	serine carboxypeptidase-like	XM_038834759.1	94%

**Table 2 genes-15-00131-t002:** Analysis of primer information and PCR amplification data on 21 candidate RGs.

Category	Genes	Primer Sequence (5′-3′) Forward	Primer Sequence (5′-3′) Reverse	Product (bp *)	R^2^ *	E * (%)
Candidate traditional house-keeping genes	*Actin7*	GTGTCATGGTTGGGATGGGT	ACGATACCGTGTTCGATGGG	150	0.9975	103
	*TUA*	ACTGGCTTCAAGTGCGGTAT	GATCGATGCGTCCAAACACC	133	0.9962	104
	*TUB*	GCCGGACAATTTCGTCTTCG	TTCCTCACGACATCGAGCAC	108	0.9988	104
	*eIF4A*	CCGGCAAGACCTCCATGATT	TTCCCTCGTAGGCGACACTA	95	1.0000	108
	*eIF5A*	GAGGAGCACCACTTCGAGTC	GGGCGATTCTTGGTGACGAT	130	0.9985	102
	*GAPDH*	GGCTTGAGAAGGAGGCTACC	ACCAGCCTTGGCGTCAAATA	158	0.9991	110
	*CYP72A*	GACGCCGAGTGTGACGATAA	GGTCCATTTCTCGCCCTCAA	142	0.9961	101
	*UBC2*	GCTCTGGAACGCTGTCATTT	CGAAGCGAACTGTAGGAGGC	116	0.9995	109
	*RAP2*	GCTTTACCGAGGAGTCAGGC	CCTCGGCGGTATCAAAGGTA	107	0.9921	105
	*UBQ*	CCCTTGAAGTGGAAAGCAGTG	ATCAGCCAGAGTCCTTCCATC	134	0.9992	104
	*ClpA*	TGCTGGAACCAAGTACCGTG	TGCCCCAGCTCCAATTAAGG	124	0.9990	105
	*FtsH2*	GGAGCTGATCTTGCCAACCT	CCCTCCATTCCAGCCACAAT	122	0.9967	104
Candidate new RGs	*RTNLB1*	CGGAGCATACTGGTGAGCAT	ATCGGAATCGGAAGACGACG	104	0.9981	98
	*UNC*	TGGTACTTCGGGTTTGCAGC	TTGATGGCGTGCGAAGGTAT	135	0.9906	104
	*GAD*	TCAGTCCACTCCACTTTCGC	TTCCCGTCCAACATCAGCTC	128	0.9988	110
	*ALEU*	GTCGGCAACACTCATAACGC	ACAGCGAGCTTGTAAGGCAA	164	0.9989	105
	*BLH1*	CCACCGCACTCCAACCTAAT	ATGAGTCCGTGCAAAGCAGA	155	0.9933	104
	*SBT3.17*	GAGGTTGACGCAATCGTTGT	ATGTGGACCTTCGATTCGGG	99	0.9994	103
	*HMGB2*	CCAAGGATCCGAACAAGCCT	CCAACAACGGCAACGGATTT	115	0.9992	103
	*GEPI48*	CTGGCATTGGTTGTGAGGTG	TTCACGTTCTGCCTTGTCCG	173	0.9942	105
	*SCPL*	GTGGGCATTCCTGCTCTTCT	GACCAGACCACTTCATGGCA	121	0.9968	103

* Here, bp, R^2^, and E denote the base pair, PCR efficiency, and correlation coefficient, respectively.

**Table 3 genes-15-00131-t003:** The rank of candidate RGs and their stability values as calculated by NormFinder.

Category	Rank	Leaves under Different Temperature Stresses		Leaves of Plants at Different Growth Stages		Different Organs	
		Gene	Stability Value	Gene	Stability Value	Gene	Stability Value
Candidate traditional housekeeping genes	1	*TUA*	0.25	*eIF5A*	0.135	*eIF5A*	0.308
	2	*eIF5A*	0.335	*TUB*	0.2	*GAPDH*	0.438
	3	*CYP72A*	0.408	*GAPDH*	0.304	*ClpA*	0.573
	4	*UBC2*	0.545	*FtsH2*	0.4	*FtsH2*	0.658
	5	*GAPDH*	0.603	*TUA*	0.411	*RAP2*	0.745
	6	*ClpA*	0.635	*ClpA*	0.685	*TUA*	0.778
	7	*TUB*	0.642	*eIF4A*	0.689	*TUB*	0.796
	8	*eIF4A*	0.682	*CYP72A*	0.748	*Actin7*	0.842
	9	*Actin7*	1.071	*Actin7*	0.778	*eIF4A*	0.924
	10	*UBQ*	1.231	*UBQ*	0.824	*UBQ*	1.019
	11	*FtsH2*	1.389	*RAP2*	0.943	*CYP72A*	1.245
	12	*RAP2*	1.418	*UBC2*	1.038	*UBC2*	1.301
Candidate new RGs	1	*SCPL*	0.102	*ALEU*	0.237	*HMGB2*	0.508
	2	*RTNLB1*	0.217	*HMGB2*	0.249	*RTNLB1*	0.518
	3	*SBT3.17*	0.292	*SCPL*	0.433	*UNC*	0.535
	4	*GEPI48*	0.373	*RTNLB1*	0.451	*GAD*	0.608
	5	*UNC*	0.467	*UNC*	0.507	*BLH1*	0.703
	6	*GAD*	0.474	*GAD*	0.531	*SCPL*	0.743
	7	*BLH1*	0.526	*BLH1*	0.55	*SBT3.17*	0.855
	8	*HMGB2*	0.7	*SBT3.17*	0.802	*GEPI48*	0.945
	9	*ALEU*	1.004	*GEPI48*	1.008	*ALEU*	1.996

**Table 4 genes-15-00131-t004:** The rank of candidate RGs and their SD and CV values as calculated by Bestkeeper.

Category	Rank	Leaves under Different Temperature Stresses			Leaves of Plants at Different Growth Stages			Different Organs		
		Gene	SD *	CV *	Gene	SD	CV	Gene	SD	CV
Candidate traditional house-keeping genes	1	*eIF5A*	0.32	1.31	*TUB*	0.33	1.29	*Actin7*	0.52	2.07
	2	*UBC2*	0.36	1.26	*GAPDH*	0.33	1.4	*eIF5A*	0.52	2.22
	3	*GAPDH*	0.36	1.37	*eIF5A*	0.34	1.53	*UBQ*	0.61	2.31
	4	*TUA*	0.45	1.61	*TUA*	0.43	1.68	*TUA*	0.67	2.59
	5	*UBQ*	0.61	2.65	*FtsH2*	0.49	2.02	*TUB*	0.82	3.11
	6	*TUB*	0.67	2.35	*ClpA*	0.5	2.48	*GAPDH*	0.85	3.32
	7	*CYP72A*	0.75	2.77	*CYP72A*	0.53	2.21	*eIF4A*	0.91	3.49
	8	*ClpA*	0.8	3.7	*UBQ*	0.53	2.5	*FtsH2*	1	3.56
	9	*RAP2*	0.9	3.16	*eIF4A*	0.57	2.3	*ClpA*	1.09	4.63
	10	*eIF4A*	0.9	3.41	*Actin7*	0.7	2.84	*RAP2*	1.25	4.4
	11	*FtsH2*	1.19	4.53	*RAP2*	0.8	2.97	*UBC2*	1.29	4.6
	12	*Actin7*	1.24	4.89	*UBC2*	0.81	3.05	*CYP72A*	1.44	5.3
Candidate new RGs	1	*HMGB2*	0.65	2.64	*HMGB2*	0.38	1.63	*SCPL*	0.5	1.9
	2	*SBT3.17*	0.74	2.89	*RTNLB1*	0.42	1.77	*SBT3.17*	0.57	2.23
	3	*SCPL*	0.79	3.02	*ALEU*	0.42	1.81	*HMGB2*	0.71	2.9
	4	*GAD*	0.84	3.27	*GAD*	0.46	1.97	*GAD*	0.77	2.89
	5	*RTNLB1*	0.91	3.55	*UNC*	0.5	1.97	*RTNLB1*	0.85	3.23
	6	*BLH1*	0.92	3.69	*BLH1*	0.56	2.39	*BLH1*	0.88	3.31
	7	*GEPI48*	1.06	4.16	*SCPL*	0.6	2.35	*UNC*	0.9	3.27
	8	*UNC*	1.11	4.11	*SBT3.17*	0.69	2.78	*GEPI48*	0.9	3.42
	9	*ALEU*	1.15	4.58	*GEPI48*	0.73	3.01	*ALEU*	1.47	5.56

* Here, SD and CV denote the standard deviation and coefficient of variation, respectively.

**Table 5 genes-15-00131-t005:** The rank of candidate RGs and their stability values as calculated by RefFinder.

Category	Rank	Leaves under Different Temperature Stresses		Leaves of Plants at Different Growth Stages		Different Organs	
		Gene	Stability Value	Gene	Stability Value	Gene	Stability Value
Candidate traditional house-keeping genes	1	*eIF5A*	1.41	*TUB*	1.41	*eIF5A*	1.68
	2	*TUA*	2	*eIF5A*	1.97	*Actin7*	2.63
	3	*UBC2*	2.38	*TUA*	2.99	*TUA*	3.13
	4	*GAPDH*	3.87	*GAPDH*	3.35	*GAPDH*	3.31
	5	*CYP72A*	4.21	*FtsH2*	4.95	*ClpA*	5.05
	6	*TUB*	6.48	*ClpA*	5.63	*TUB*	5.21
	7	*ClpA*	7.44	*eIF4A*	5.8	*FtsH2*	6.16
	8	*eIF4A*	7.61	*CYP72A*	7.97	*UBQ*	6.51
	9	*UBQ*	8.61	*Actin7*	8.97	*eIF4A*	7.94
	10	*Actin7*	9.67	*UBQ*	9.69	*RAP2*	7.95
	11	*FtsH2*	10.74	*RAP2*	10.74	*CYP72A*	11.24
	12	*RAP2*	11.17	*UBC2*	12	*UBC2*	11.74
Candidate new RGs	1	*SCPL*	1.86	*ALEU*	1.86	*HMGB2*	1.73
	2	*RTNLB1*	2.78	*RTNLB1*	2	*SCPL*	2.21
	3	*GEPI48*	3.03	*HMGB2*	2.34	*SBT3.17*	3.15
	4	*SBT3.17*	3.31	*GAD*	3.31	*RTNLB1*	3.5
	5	*UNC*	3.76	*UNC*	4.16	*UNC*	3.6
	6	*HMGB2*	4.76	*SCPL*	5.45	*GAD*	4.68
	7	*GAD*	5.42	*BLH1*	6.48	*BLH1*	5.96
	8	*BLH1*	6.74	*SBT3.17*	8	*GEPI48*	8
	9	*ALEU*	9	*GEPI48*	9	*ALEU*	9

## Data Availability

No new data were created or analyzed in this study. Data sharing is not applicable to this article.

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
