# Peer review of "Selection and Validation of Reference Genes for Gene Expression Studies in *Euonymus japonicus* Based on RNA Sequencing"

_genes, 2024, doi:10.3390/genes15010131_

Round 1
Reviewer 1 Report
Comments and Suggestions for Authors
The manuscript presents the expression stability pattern obtained from RNA-seq and RT-qPCR used to propose suitable reference genes for future expression and regulation data normalization of Euonymus japonicus under specific conditions. The subject is of interesting and may contribute with future gene expression studies in this species. However, experimental design description is superficial and unreproducible. Thus, the points highlighted bellow must be sufficiently described and justified.
2. Material and methods
2.1 Plant material
How leaves age was determined?
Change tissue by organs (page 2, line 88)
2.2 RNA-seq
This topic must be improved with essential details such as: library preparation, sequencing mode, fragment length, Program assembler versions and their respective citations. What type of transcriptome was sequenced? How many replicates were included?
How the most stable unigenes were identified from the large RNA-seq dataset? What stability criterion was used?
2.4. Primer design and PCR
How primer specificity was insured? I mean, many genes selected in this study are members of a multigene family, which possess closely related genes. Otherwise, qPCR results are uncertain and unreliable.
How amplification specificity was insured? DNAase treatment was employed? Primers designed overlapped at least an exon/exon junction? Otherwise, qPCR results are uncertain and unreliable.
Authors do not indicate if biological and/or technical replicates were considered?
2.5 Data analysis
In the sentence “are used to analyze the expression stability of candidate RGs” were suite better than are.
3. Results
RNA-seq FPKM values should be shown as mean +/_ CV and included as supplementary table. I missed RNA-seq data presentation.
Figure 3. Melting curves of 9 candidate reference genes. Would not be 21 genes?
3.4. Stability Evaluation of Candidate Gene
The related paragraph is not “results”, it sounds redundant because it was described in the material and methods section.
Authors should clarify why they selected EjMAH1 for confirmation? I mean, explain why this gene is a reliable marker? What is the function of EjMAH1?
5. Conclusion
Authors analyzed and ranked traditional and novel housekeeping genes separated. Further, showed the best combinations inside each group. Nevertheless, they conclude that eIF5A and RTNLB1 can perform better. I missed to see the validation combining of two best traditional and novel housekeeping genes. I mean, in practice, how eIF5A and RTNLB1 would perform in normalization of EjMAH1
Reviewer 2 Report
Comments and Suggestions for Authors
The manuscript “Selection and validation of reference genes for gene expression studies in Euonymus japonicus based on RNA-seq” described the criteria and selection of reference genes that are suitable for further RT-PCR reaction based on the transcriptomic data. As a result, eIF5A and RTNLB1 were identified and verified to act as the suitable RGs of Euonymus japonicus in this experiment.
Other suggestions
What are the reasons to choose the structural gene EjMAH1 as a target and examine its expression pattern instead of using the temperature sensitive genes as a test and verification for the newly identified internal control? What is the functionality of this gene in response to temperature stress? More explanations are needed.
Comments on the Quality of English LanguageMinor errors
Page 3, Line 100, assess, spelling error instead of assesse
Page 3, Line 106, total RNA instead of RAN, spelling error
Page 4, Line 141, homologous sequence, spelling error
Page 9, Line 267, The figure before the discussion should be figure 6, wrong label in the figure legend, please correct that
Reviewer 3 Report
Comments and Suggestions for Authors
Manuscript "Selection and validation of reference genes for gene expression studies in Euonymus japonicus based on RNA-seq" is very interesting.
General comments:
Authors used leaves of E. japonicus at different overwintering times as materials and selected 11 candidate traditional housekeeping genes and nine candidate new RGs were based on the results of RNA-seq.
Authors estimated of expression stability using geNorm, NormFinder, Bestkeeper and RefFinder.
Detailed comments:
The introduction was well written. The background was outlined and the research problem highlighted.
The description of materials and methods is quite detailed, except for subsection "2.5. Data analysis." In this subsection, the authors only listed the tools, without providing specific methods. The description of the methodology should make it possible to repeat the experiment. Unfortunately, the current laconic description does not make this possible. This section should be supplemented with a detailed description of the research methods used.
Table 2: E (%) - should be clarified.
Figure 3: too laconic signature.
Figure 4: statistical comparative analysis between profiles is missing.
Table 3: A comparative scale for stability factors is missing.
Table 4: All abbreviations used should be explained.
Table 5: A comparative scale for stability factors is missing.
Figure 5: should be supplemented with the results of the two-way analysis of variance.
Paper needs major revision.
Round 2
Reviewer 1 Report
Comments and Suggestions for Authors
All my major concerns were met in this revised version.
Reviewer 3 Report
Comments and Suggestions for Authors
The authors have significantly improved the manuscript. The responses to my comments and feedback are satisfactory. I support publication of the manuscript in its present form.